# Autonomous self-healing supramolecular polymer transistors for skin electronics

Ngoc Thanh Phuong Vo[1,6], Tae Uk Nam[1,6], Min Woo Jeong[1], Jun Su Kim[1], Kyu Ho Jung[1], Yeongjun Lee[2,3], Guorong Ma[4], Xiaodan Gu[4], Jeffrey B.-H. Tok[2], Tae Il Lee[5] ✉, Zhenan Bao[2] ✉ & Jin Young Oh[1] ✉

Skin-like field-effect transistors are key elements of bio-integrated devices for future user-interactive electronic-skin applications. Despite recent rapid developments in skin-like stretchable transistors, imparting self-healing ability while maintaining necessary electrical performance to these transistors remains a challenge. Herein, we describe a stretchable polymer transistor capable of autonomous self-healing. The active material consists of a blend of an electrically insulating supramolecular polymer with either semiconducting polymers or vapor-deposited metal nanoclusters. A key feature is to employ the same supramolecular self-healing polymer matrix for all active layers, i.e., conductor/semiconductor/dielectric layers, in the skin-like transistor. This provides adhesion and intimate contact between layers, which facilitates effective charge injection and transport under strain after self-healing. Finally, we fabricate skin-like self-healing circuits, including NAND and NOR gates and inverters, both of which are critical components of arithmetic logic units. This work greatly advances practical self-healing skin electronics.

Skin-inspired thin-film field-effect transistors are core elements of integrated circuits for electronic skin to integrate with the human body. They have rapidly advanced the development of stretchable electronic materials[1–3]. They have demonstrated strong potential for applications in health monitoring, prosthetic sensory skin, medical implants, and brain-computer interface[4,5]. Despite substantial progress of the above devices, studies on self-healing ability that is critical for skin-like devices when subjected to unexpected mechanical damages remain lacking[6].

Intrinsically self-healing insulating materials have been developed with supramolecular polymer chemistry and are capable of reconstruction through dynamic intermolecular interactions[7–9]. However, for electronic materials, most self-healing polymers reported to date are electrically insulating, which limits their applications in functional self-healing transistors[7]. Thus, self-healing semiconductors and electrodes are ideal for skin-inspired transistor applications[6,10–12]. Self-healing conductors have been developed using composite materials with conducting nanofillers, such as carbon materials and metal nanowires or liquid metals[13–18]. However, self-healing semiconductors are less developed. In addition, the above materials have yet been thoroughly investigated for use in self-healing field-effect transistors. Consequently, realizing fully autonomous self-healing transistors have the following five major challenges[19]: (i) low elasticity of the self-healing semiconductor that induces fatigue failure, (ii) non-autonomous healing process involving heat and solvent treatments are unfavorable for practical applications, (iii) difficult to align limited healing area (submicron scale), (iv) strain-sensitive electrical properties preventing stable electrical characteristics, and (v) absence of

[1]Department of Chemical Engineering (Integrated Engineering Program), Kyung Hee University, Yongin, Gyeonggi 17104, Korea. [2]Department of Chemical Engineering, Stanford University, Stanford, CA 94305-5025, USA. [3]Department of Brain and Cognitive Sciences, KAIST, Daejeon 34141, Korea. [4]School of Polymer Science and Engineering, University of Southern Mississippi, Hattiesburg, MS 39406, USA. [5]Department of Materials Science and Engineering, Gachon University, Seong-nam, Gyeonggi 13120, Korea. [6]These authors contributed equally: Ngoc Thanh Phuong Vo, Tae Uk Nam. ✉e-mail: t2.lee77@gachon.ac.kr; zbao@stanford.edu; jyoh@khu.ac.kr

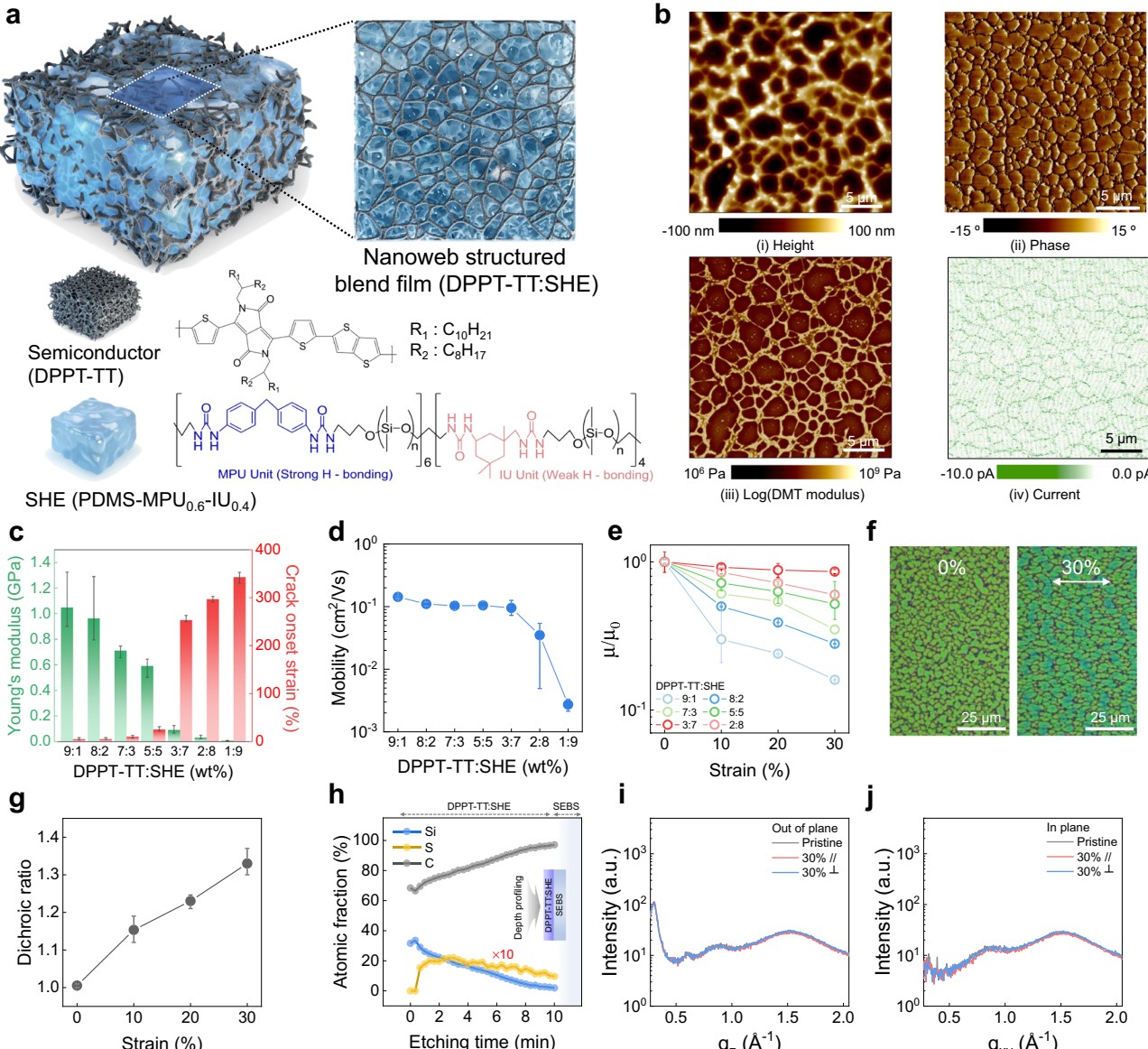

**Fig. 1 | Material design of autonomous self-healing semiconductor. a** Schematic of an autonomous self-healing semiconducting film, featuring the chemical structure of a self-healing elastomer (PDMS−MPU$_{0.6}$−IU$_{0.4}$) and a semiconducting polymer (DPPT-TT). **b** AFM analysis of the 3:7 (DPPT-TT: SHE) blend ratio film: (i) height, (ii) phase, (iii) DMT modulus, and (iv) C-AFM images. **c** Elastic modulus and onset strains of cracks in the films with different blend ratios of self-healing semiconductor. **d** Field-effect mobilities of blend films as a function of the blend weight ratio (DPPT-TT:SHE). **e** Field-effect mobility ratios of the blend films as a function of the blend weight ratio and tensile strain. **f** OM images of the 3:7 blend ratio film at 0% (left) and 100% strain (right). **g** Dichroic ratio from polarized UV−Vis−NIR spectra at each tensile strain. **h** XPS spectra of the 3:7 blend film. GIXD of the 3:7 blend film under 0% and 30% strain, where the strained film was stretched parallel and perpendicular to the beam line. **i** out-of-plane and (**j**), in-plane film direction. All error bars were calculated using a sample in each of three batches (*n* = 3).

suitable self-healing electrode and dielectric materials for both efficient current injection and a low operating voltage of the transistors[10–12,20–22]. In this work, we address these challenges by introducing a supramolecular polymer transistor consisting of identical supramolecular self-healing polymers in conductor, semiconductor, and dielectric[22,23]. Such transistors can autonomously self-heal of micron-scale damage (up to 4 µm) in transistor configuration at ambient conditions and operate at a low drain voltage (−1 V). Using our self-healing transistors, we proceed to fabricate an active-matrix array and logic gates. Their performance was maintained even when subjected to 30% biaxial strain. This work represents the first report of an autonomously self-healing integrated logic circuit, showing the potential to incorporate stretchable self-healing functions into future, more complex skin electronics.

## Results

### Material design of autonomous self-healing semiconductor

Figure 1a shows a schematic of a self-healing semiconducting film based on a blend of a semiconducting polymer and a supramolecular elastomer. We used poly(2,5-bis(2-octyldodecyl)−3,6-di(thiophen-2-yl) diketopyrrolo[3,4-c]pyrrole-1,4-dione-*alt*-thieno[3,2-*b*]thiophene) (DPPT-TT) as the semiconductor and poly(dimethylsiloxane) (PDMS)−4,4′-methylenebis(phenyl urea) (MPU)−isophorone bisurea (IU) (PDMS−MPU$_{0.6}$−IU$_{0.4}$) as the supramolecular elastomeric matrix (self-healing elastomer, SHE) (Fig. 1a). The DPPT-TT semiconducting polymer phase-separated into nanoconfined fibers because of strong π−π interactions between DPPT-TT molecules. The nanoconfinement effect of polymer semiconductors in blends was previously reported to reduce conformational disorder and resulted in sometimes

even enhanced mobility in stretchable semiconducting films, despite inclusion of a substantial amount of insulating material[24,25]. PDMS−MPU$_{0.6}$−IU$_{0.4}$ is a previously reported self-healing polymer consisting of dual-strength dynamic H-bonding sites[21]. Such a molecular design enables efficient energy dissipation in response to strain and provides elasticity as well as a high fracture energy.

The blended semiconductor film was observed to have a nanoweb-like network of the semiconducting polymer that provides geometrical stretchability, while maintaining current flow pathways with high interconnectivity within the semiconducting polymer phase (Fig. 1b(i) and Fig. 1b(ii)). The nanoweb morphology was observed in as-spun blend film, which was a result of phase separation due to the difference in surface energy between the materials, and the nanoweb morphology in the SHE matrix maintained after thermal annealing at 80 °C for 30 min (Supplementary Figs. 1, 2). Figure 1b(iii) shows the Derjaguin−Muller−Toporov (DMT) modulus mapping image of the semiconductor film, which indicates that the nanoweb network is composed primarily of the semiconducting polymer (DPPT-TT: $10^2$ MPa level, SHE: $10^1$ MPa). The percolating pathways for current transport in the semiconducting nanoweb were confirmed using conductive AFM (C-AFM) (Fig. 1b(iv)). The weight ratio of DPPT-TT to SHE was varied to control the Young's modulus and the crack onset strain of the blend films (refer to details in Supplementary Note 2). A tradeoff relationship was observed between these two factors (Fig. 1c). The elastic modulus dramatically decreased and the crack onset strain substantially increased when the DPPT-TT:SHE weight ratio of the blend film was 3:7 (Fig. 1c and Supplementary Figs. 3–5). To gain insight into the mechanical properties of the 3:7 blend film, dynamic mechanical analysis (DMA) was conducted on the blend film (DPPT-TT:SHE, 3:7). The storage modulus (G′) consistently exceeded the loss modulus (G″) across the $10^{-1}$–$10^1$ Hz frequency range at 35 °C, and the difference between the storage and loss moduli became larger as the frequency increased. This indicates that the blend film exhibits solid-like viscoelastic properties (tan δ = G′/G″ <1) in room temperature (Supplementary Fig. 6a). In addition, the temperature dependency of the tan δ was investigated from −20 °C to 60 °C at 1 Hz and the tan δ was remained below 1 in the temperature range, which means that the blend film behaves like a solid-like material (Supplementary Fig. 6b). The unchanged length of the blend film suspended under the gravity for 7 days at room temperature supports its solid-like behavior (Supplementary Fig. 7). This ratio was also observed as a transitional point for the mechanical and electrical properties of the blended film (Fig. 1b and Supplementary Fig. 8).

The field-effect mobility was evaluated at weight ratios ranging from 1:9 to 9:1 (DPPT-TT:SHE), as shown in Fig. 1d and Supplementary Figs. 9 and 10. The measured mobility changed only slightly, indicating that the electrical percolation path remained almost intact in films with DPPT-TT:SHE weight ratios between 9:1 and 3:7. The strain dependency of the semiconducting property was further evaluated through transfer printing of the blend film (Supplementary Figs. 11, 12). The field-effect mobilities of the selectively stretched blend films on a rigid substrate were measured (Fig. 1e). The blended film with a weight ratio of 3:7 exhibited the lowest sensitivity toward strain in terms of field-effect mobility. Optical microscopy (OM) images clearly showed the strain-insensitive nanoweb structure of the 3:7 blended film, i.e., with no observable cracks even after 30% strain was applied (Fig. 1f and Supplementary Fig. 4). Consequently, the four kinds of material factors (nanoweb morphology, strain-insensitive electrical property, crack-onset strain, and Young's modulus) were considered for the optimal blend ratio of the semiconducting film.

We used the dichroic ratio to determine the degree of relative DPPT-TT polymer alignment in the blended films under strain[26]. The dichroic ratio of the 3:7 blend film increased linearly as the strain was increased to 30% (Fig. 1g and Supplementary Fig. 13), suggesting that the DPPT-TT chains were aligned in the strain direction. To gain further

insights into the properties of the 3:7 blend film, we carried out depth profiling on three unique points of the film using X-ray photoelectron spectroscopy (XPS) to analyze the vertical composition distribution. All component distribution (S, Si, and C atoms) in depth of the three points all showed identical trend and the average S and Si atoms were representative of DPPT-TT and SHE, respectively (Fig. 1h and Supplementary Fig. 14). The S signal was evenly distributed throughout the film thickness, suggesting that DPPT-TT (the only source of S signals) was distributed throughout the film. This even distribution potentially provided continuous pathways for charge injection and charge transport in the top-contact field-effect transistor. Si atoms from the self-healing PDMS polymer were found in a higher concentration than S atoms near the surface, indicating that the PDMS polymer may have encapsulated the active channel region at the semiconductor-dielectric interface. The 3:7 blend film was further analyzed by grazing-incidence X-ray diffraction (GIXD) (Fig. 1i, j and Supplementary Figs. 15, 16). Its GIXD pattern was almost unchanged under strain as high as 30%, irrespective of the stretching direction. This result suggests that the semiconductor's nanoweb structure helped to release the applied strain and preserved the crystalline domain. This feature is useful to reduce the strain effects on the film's charge transport properties.

## Autonomous self-healing semiconductor
We next conducted self-healing test on the blended film (weight ratio 3:7) (Fig. 2a). A semiconducting film on a self-healing dielectric layer was hand cut to have a width of 2.5 μm, a depth of 0.1 μm and a centimeter-scale length via OM images (Fig. 2b, left and Supplementary Fig. 17). The damaged films were observed to be gradually filled by surrounding materials, leading to complete healing of the damage after 30 h at room temperature (Fig. 2b, right). The microscale damaged areas were characterized by AFM measurements (Fig. 2c–e and Supplementary Fig. 18), and the width of the damaged zone was barely visible after self-healing, while the remaining depth was ~10 nm (Fig. 2e, bottom). With a damaged gap width of 4 μm, we observed that the blended film on dielectric was able to heal the cut (Fig. 2f and Supplementary Figs. 19–22). In addition, the scattering profiles before and after healing were almost identical (Supplementary Fig. 23) and the autonomous self-healing of the blend film was found to be repeatable, even if the healed region is damaged again (Supplementary Fig. 24). However, when the damaged gap width was wider, i.e., 5 μm, the damaged region was unable to undergo complete self-repair, but the gap width of the damaged region was reduced by over 50% (from 5 μm to ~2 μm) (Supplementary Fig. 25). In case of such larger width of damages, self-healing of the damaged semiconducting film could be achieved only through post-treatments such as exposure to heat or solvent vapor[10]. These treatments facilitated more movement of the self-healing elastomer, allowing it to fill the damaged gap up to ~9 μm and enabling quick reconnection of the DPPT-TT network in the elastomer matrix within 10 min (Supplementary Figs. 26–28).

With these results, we next fabricated thin-film field-effect transistors with a self-healing semiconducting/dielectric film on an indium tin oxide (ITO) (gate)/glass rigid substrate (process detailed in Supplementary Fig. 29). The self-healing properties of the semiconducting film were then evaluated. The transfer curves of the transistors before, during, and after self-healing are shown in Fig. 2g–j. The pristine transistor displayed a typical transfer curve and an average field-effect mobility of 0.12 cm$^2$/(V·s). The semiconductor was again cut in between the source and drain, perpendicular to the current path, using a surgical blade; the damaged gap width was 3 μm, and the depth was 0.1 μm, which is the semiconductor thickness (Fig. 2g and Supplementary Fig. 29a). The damaged device worked again after 12 h, and the mobility gradually increased with increasing healing time. After 3 days, the mobility was restored back akin to an intact device. In addition, the transfer curve of a transistor with a smaller damage width of

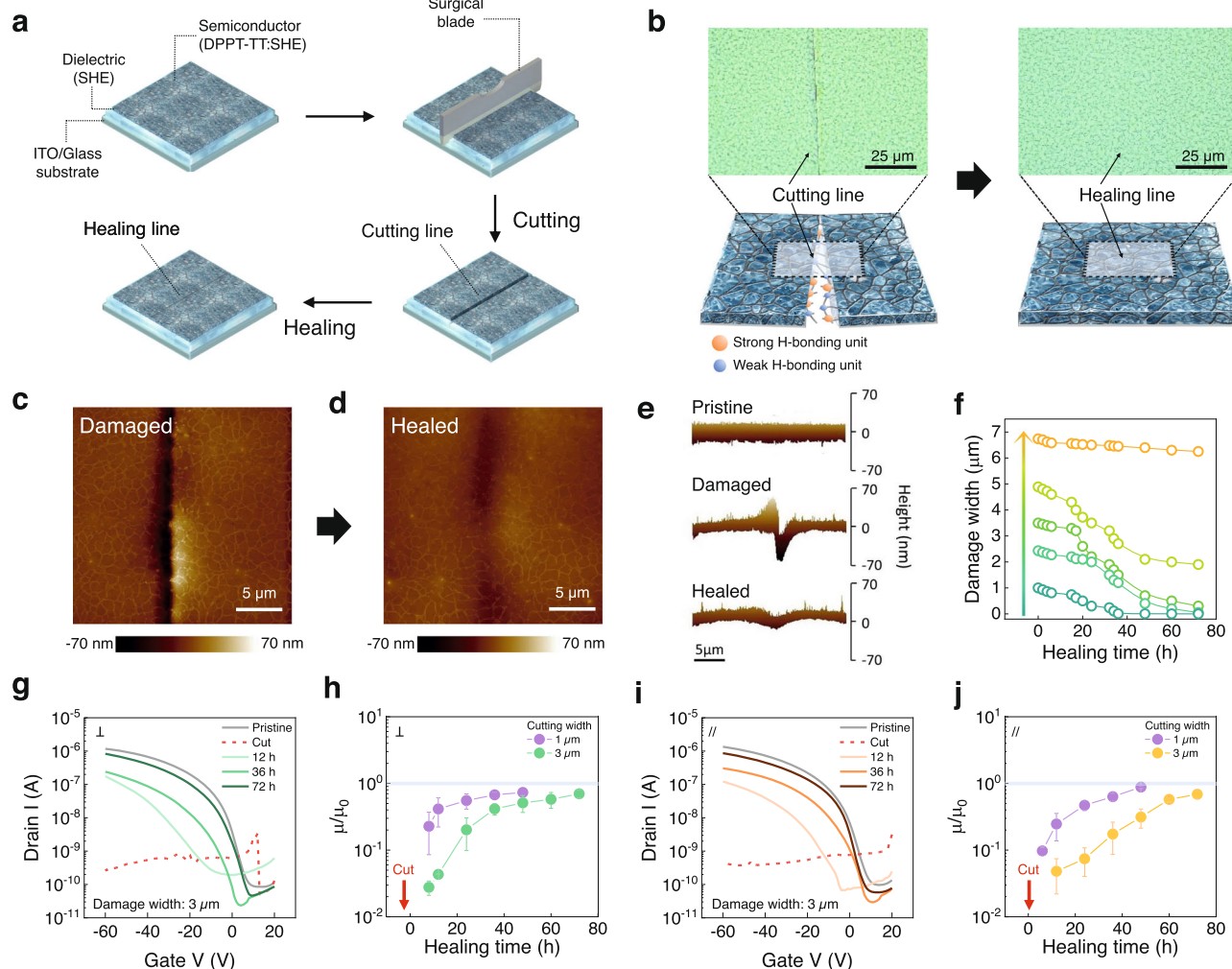

**Fig. 2 | Characterization of autonomous self-healing semiconductor film.**
**a** Illustration of the self-healing process. **b** OM images and schematic of a semi-conducting film cut using a surgical blade (left) and the film after healing (right). AFM height image of a semiconducting film **c** cut with a surgical blade and (**d**), after healing. **e** Change in the height profile of a self-healing semiconducting film for the self-healing process. **f** Self-healing ability of a semiconducting film, plotted as the change in the damage width as a function of time for different damage cutting sizes. **g** Transfer characteristics ($V_D = -60$ V) with a cutting line (3 μm width)

perpendicular to the channel as a function of the healing time. **h** Normalized field-effect mobilities ($\mu_0$: pristine mobility) with different cutting widths for transistors subjected to cutting perpendicular to the channel direction as a function of healing time ($n = 3$). **i** Transfer characteristics ($V_D = -60$ V) with a cutting line (3 μm) parallel to the channel as a function of healing time. **j** Normalized field-effect mobilities ($\mu_0$: pristine mobility) with different cutting widths parallel to the channel direction as a function of the healing time ($n = 3$). All error bars were calculated using a sample in each of three batches ($n = 3$).

1 μm, measured directly after cutting, is shown in Supplementary Fig. 29c. We observed that electrical healing was faster when the damage width is narrower (1 μm vs 3 μm; Fig. 2h). A fast recovery of the transistor characteristics to their initial state was after 12 h. When the current path of the semiconductor was cut in parallel (Supplementary Fig. 29b), autonomous self-healing occurred at similar healing times (Fig. 2i, j and Supplementary Fig. 29d).

**Autonomous self-healing electrodes**

Self-healing electrodes and dielectrics have been investigated for fully autonomous self-healing transistors[18,27–29]. Here, we used a previously reported stretchable metallization approach for self-healing electrodes, where vaporized Ag atoms form a metal-polymer nanocomposite on the surface of elastic semiconducting and dielectric films, resulting in robust stretchable electrodes compared to other noble metals (Au and Cu)[23]. In addition, the native oxide of Ag ($Ag_xO$) with high work function ($\Phi_W = 5.2$ eV) in the nanocomposite allows good contact with DPPT-TT semiconductor (HOMO level = 5.04 eV) for stretchable transistors[23]. To verify the compatibility of Ag metallization for

self-healing transistors, we evaluated the self-healing performance of the Ag metallized self-healing semiconductor (as source/drain electrodes) and self-healing substrate (as a gate electrode). The Ag-supramolecular polymer nanocomposite was confirmed by XPS depth profiling (Fig. 3a), and the $Ag_xO$ was observed in the nanocomposite (Fig. 3b, c). Consequently, the effective Schottky barrier height for current injection was below 0.1 eV (Fig. 3d and Supplementary Fig. 30)[23,30]. Figure 3e shows the variations in resistance of the Ag-metallized self-healing gate electrode under strain. The initial resistance of the Ag gate electrode was 13 Ω (Ag thickness: 80 nm). The resistance of the Ag electrode on the SHE films was found to be stable under biaxial strain as high as 30% and remained <100 Ω after 100 cycles under 30% biaxial strain, based on nano-crack damages (Fig. 3f and Supplementary Fig. 31a, c). The self-healing property of the electrode was investigated through changes in the electrode resistance with different damage size as a function of self-healing time (Fig. 3g). After the electrodes were cut, the resistance of the electrode with a 3 μm damage width increased dramatically to the $10^7$ Ω level, mostly electrically disconnected. However, the resistance recovered after

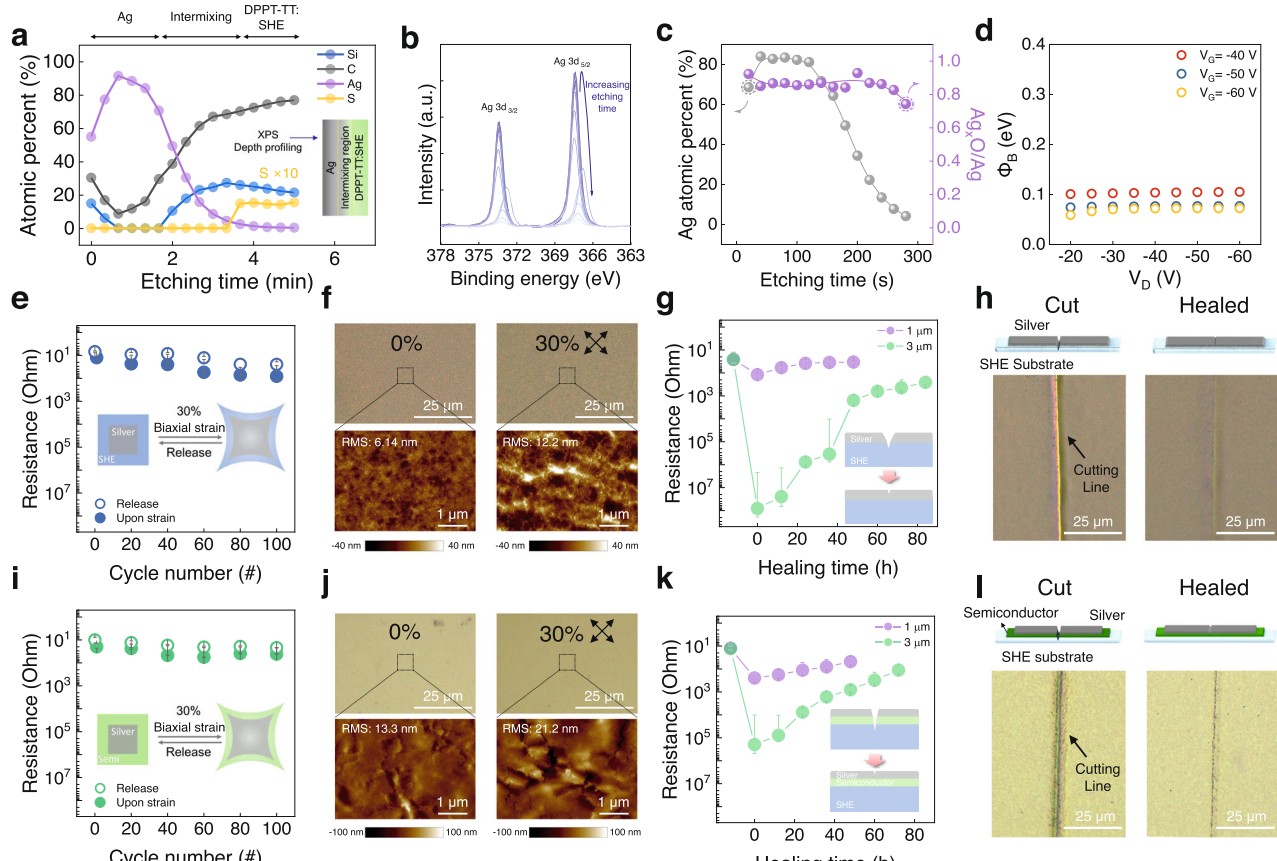

**Fig. 3 | Autonomous self-healing electrodes. a** XPS depth profiling of Ag metallized DPPT-TT:SHE film, (**b**) XPS spectra of Ag 3d peak, and (**c**) $Ag_xO/Ag$ ratio as function of etching time. **d**. Effective Schottky barrier height of Ag/DPPT-TT:SHE. **e** Resistance change ($n = 3$) of a gate electrode during 100 cycles at 30% biaxial strain (inset image shows a biaxially stretched gate electrode on the SHE film). **f** OM and AFM images of the gate electrode at 0% (left) and 30% biaxial strain (right). **g** Self-healing ability of the gate electrode as a function of the healing time for different cutting sizes (inset image shows an illustration of a self-healing gate electrode on the SHE film). **h** Illustrations (top) and OM images (bottom) of a gate electrode in the cut (left) and healed (right) states. **i** Resistance change ($n = 3$) of the electrode on a self-healing semiconductor (DPPT-TT:SHE) during 100 cycles at 30% biaxial strain (inset image shows a biaxially stretched electrode on a self-healing semiconductor). **j** OM and AFM images of an electrode on a semiconductor at 0% (left) and 30% biaxial strain (right). **k** Self-healing ability of an electrode on a self-healing semiconductor as a function of the healing time with different cutting sizes. **l** Illustrations (top) and OM images (bottom) of an electrode on a semiconductor: cut (left) and healed (right) states. All error bars were calculated using a sample in each of three batches ($n = 3$).

1 day and almost reached its original value after 80 h. The rapid initial self-healing process is attributed to the elastic recovery of the supramolecular elastomer near the damaged area. However, it is followed by a slower electrical recovery (after 40 h), which may be due to surface rearrangement and diffusion of the polymer chains[31,32]. For the narrower damage width at 1 μm, the resistance was initially reduced to the $10^2 Ω$ level and fully recovered to its initial value after 24 h. Additionally, the cutting lines in the optical images of the cut electrode almost disappeared after 24 h (Fig. 3h and Supplementary Figs. 32, 33). The reconstruction of the Ag electrodes was attributed mainly to the metal–polymer nanocomposite region composed of thermally evaporated Ag nanoclusters and mobile chains of the supramolecular polymer on the surface of the SHE film (Supplementary Fig. 34)[18,23,29]. The Ag metallization applied to the self-healing semiconductor layer also showed a stable resistance (Fig. 3i) and remained mechanically robust even when subjected to 30% biaxial strain for 100 cycles (Fig. 3j and Supplementary Fig. 31b, d). The initial resistance of the Ag electrode on the semiconducting films was $10^1 Ω$. However, when the films were cut, the resistance increased dramatically to $10^2 Ω$ and $10^4 Ω$ for both damage widths of 1 μm and 3 μm, respectively. However, the damaged Ag/semiconductor film was recovered through the reconstruction of the metal-polymer nanocomposite and the semiconductor film. As a result, the resistance of Ag metallized layer recovered after

36 h (Fig. 3k) and the cutting line almost disappeared for both cutting widths with increasing self-healing time (Fig. 3l and Supplementary Figs. 35, 36). The self-healed Ag/SHE and Ag/semiconductor films could be stretched again to 30% strain without electrical disconnection (Supplementary Fig. 37). Moreover, the metal-polymer nanocomposite layer functions as a physical glue at the interfaces (Ag/SHE and Ag/semiconductor), providing strong adhesion at the interfaces and thereby improving the mechanical robustness of the electrodes. The electrodes almost maintained their electrical resistance even after several peel-off tests using 3M tape (Supplementary Fig. 38).

### Autonomous self-healing dielectric and transistors

The self-healing capability of the dielectric layer is another critical aspect of fully stretchable self-healing transistors[33]. The supramolecular elastomer ($PDMS–MPU_{0.6}–IU_{0.4}$) was used as the stretchable and self-healing dielectric in a metal–insulator–metal (MIM) configuration of Ag/dielectric/Ag (Supplementary Fig. 39). In transistors, the dielectric strength of the gate dielectric plays a critical role in preventing electrical breakdown[34]. This self-healing dielectric (thickness: 1.5 nm) with smooth surface (roughness: <1 nm) exhibited a breakdown voltage of 170 V at a current density of $10^{-7} A/cm^2$ (Fig. 4a and Supplementary Fig. 40). Although a thinner dielectric layer can lead to a lower operating voltage in transistors, dielectric layers with a

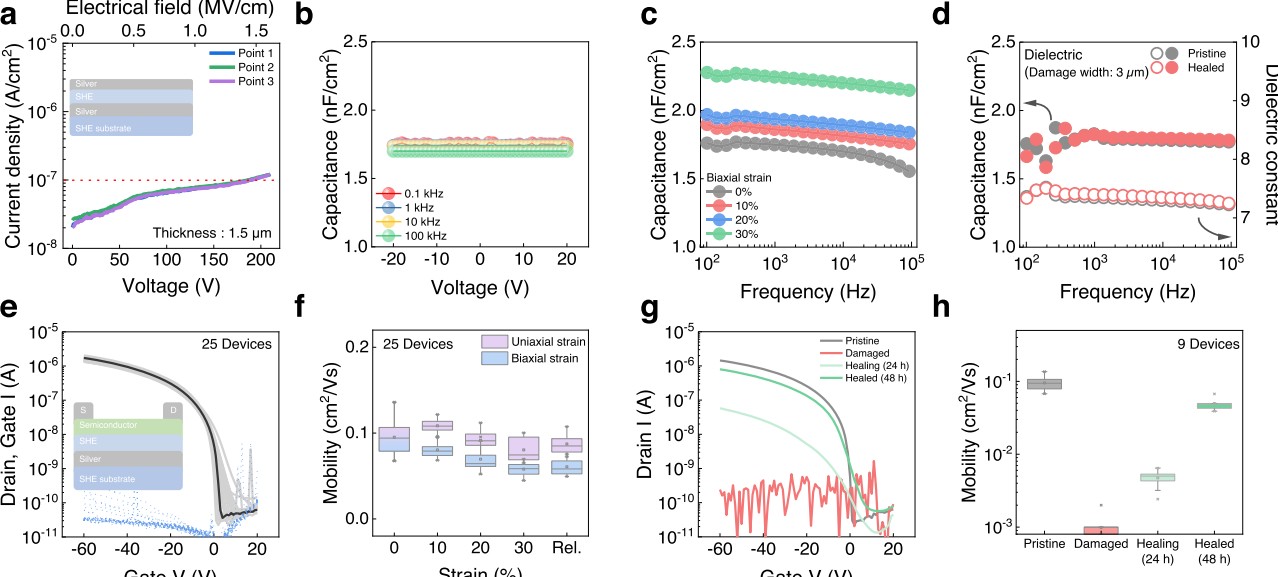

**Fig. 4 | Autonomous self-healing dielectric and transistor. a** Current density versus electrical field plots of the self-healing dielectric in MIM structure. **b** C–V performance of the self-healing dielectric at different frequencies. **c** C–F performance of a biaxially stretched dielectric at 0 V. **d** Capacitance and dielectric constant of a self-healing dielectric in its pristine and healed (3 μm cutting width) states. **e** Transfer characteristics for 25 devices of a self-healing transistor. The black line and blue dots show the drain and gate currents, respectively. **f** Mobility change under uniaxial and biaxial strain of the self-healing transistor. **g** Transfer characteristics and (**h**), mobility change of nine devices in the damaged, healing (24 h), and healed (48 h) states. All error bars were calculated using a sample in each of three batches ($n = 3$).

thickness of less than 1.5 μm resulted in a lower breakdown voltage and higher leakage current in the devices (Supplementary Fig. 41a–d). Additionally, thinner dielectric layer faces a challenge in aligning healing areas at the submicron scale, which leads to a reduced self-healing ability, even though the contact resistance and current injection did not affect the devices (Supplementary Figs. 41e–h and 42). The dielectric exhibited a consistent capacitance value in the investigated frequency range ($10^2$–$10^5$ Hz) with −20 V to 20 V (Fig. 4b)[35]; the dielectric constant (k) was 7.2. Under a biaxial strain as high as 30%, the dielectric capacitance slightly increased across the entire investigated frequency range (Fig. 4c)[36]. With a decrease in the dielectric thickness due to biaxial strain, the corresponding capacitance ($C_i$) values of the MIM capacitors at $10^3$ Hz increased from 1.74 nF/cm² (pristine, 0%) to 2.24 nF/cm² (30% biaxial strain) without any physical defects (Supplementary Fig. 40). To demonstrate the self-healing ability of the dielectric layer, damage with a width of 3 μm and length of a few centimeters was induced on the dielectric using a surgical blade and the healing time was monitored. Figure 4d shows the capacitance and dielectric constant values of the dielectric in its pristine and posthealing states (Supplementary Fig. 43).

Based on the above results, the self-healing semiconductor, conductor, and dielectric materials were successfully integrated into a single transistor device and passive arrays of 5 × 5 self-healing transistors were fabricated (Supplementary Fig. 44). Figure 4e shows typical transfer curves for 25 devices in a transistor array with a low leakage current and an on/off ratio greater than $10^4$. Typical output curves of the device without s-shape were shown, which indicates good electrical contacts at the source/drain and semiconductor interfaces (Supplementary Fig. 45a)[23,37]. The devices exhibited an anticlockwise hysteresis that is attributed to charge carrier trapping near the channel and semiconductor-dielectric interface[38]. The degree of electrical hysteresis was quantified by the shift in threshold voltage ($\Delta V_{th}$), which depends on the I·V sweep speed (Supplementary Fig. 46). All unit devices in the arrays exhibited uniform saturation mobility values (average of 25 devices: $0.11 \pm 0.02$ cm²/V·s) (Supplementary Fig. 45b) and three batches of the array ($n = 3$) showed a similar trend

(Supplementary Fig. 45c). The autonomous self-healing supramolecular polymer transistor showed temperature-dependent on-current based on hopping transport and operated up to 100 °C with maintaining its switching behavior (Supplementary Fig. 47).

For stretchability, we tested the devices under both uniaxial and biaxial strains. The cyclic strain–stress curve of the transistor array (Supplementary Fig. 48) for the strains ranging from 10% to 100% showed that the device can be stretched without permanent plastic deformations. All devices maintained their initial mobility and oncurrent under uniaxial and biaxial strains as high as 30% (Fig. 4f and Supplementary Figs. 49–52). Changes in the capacitances of the stretchable self-healing dielectrics and device geometries according to the applied strain are summarized in Supplementary Table 1.

To evaluate the self-healing efficiency, we first cut all components within the transistor array (width: 4 μm and depth: 2 μm) using a surgical blade in both directions parallel and perpendicular to the electrical pathways (i.e., forming a cross shape between the transistors), and then proceed to monitor their performance and structural recovery over time. As expected, the transistors showed complete disconnection of the pathway immediately after the array was cut. However, all the transistors were able to autonomously self-heal over time (duration dependent on cut severity), and eventually restored most of their and electrical properties and original morphology (Fig. 4g, h, Supplementary Figs. 53–56). This represents the first report that all layers in the polymer transistor are fully stretchable and autonomous self-healing (Supplementary Table 2). The healing time was observed to be the same regardless of the cut directions. Furthermore, all the transistors demonstrated >77% recovery efficiency (average mobility: $0.07 \pm 0.02$ cm²/V·s, with a maximum of 0.09 cm²/V·s). To assess the frequency response of the healed device, capacitance versus voltage (C-V) curves of the self-healing metalinsulator-semiconductor (MIS) capacitor were measured across different frequencies. The pristine MIS capacitor exhibited typical C-V curves varying with frequency. After healing, a slight hump was observed in the C-V curves between 0 V and 10 V, indicative of increased charge traps at the semiconductor/dielectric interface

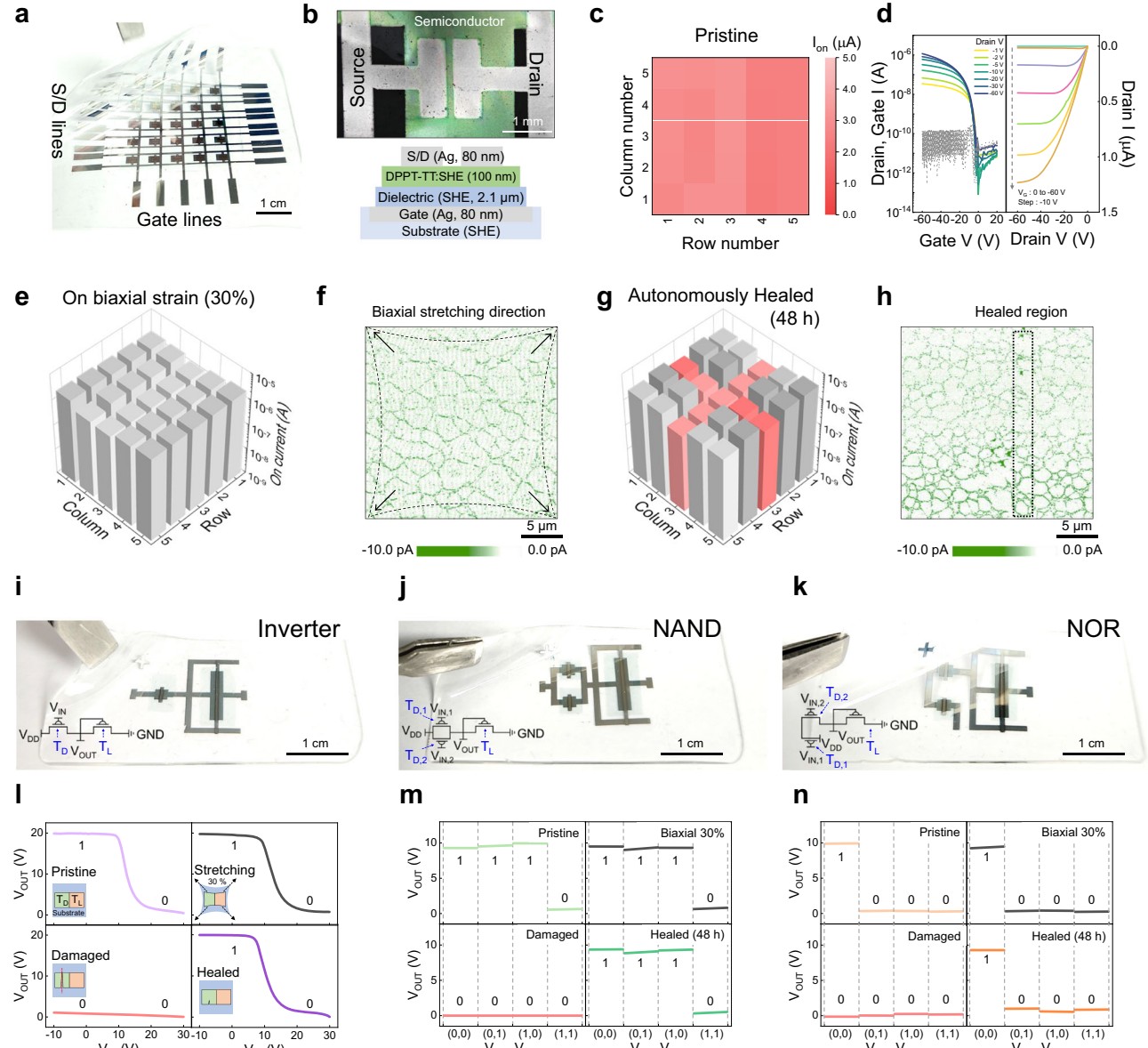

**Fig. 5 | Self-healable skin electronics. a** Photograph of self-healing-transistor active-matrix arrays. **b** OM image and structure of a unit device of the array. **c** On-current mapping ($V_D$, $V_G = -60$ V) of a pristine array. **d** Transfer characteristics with different drain voltages (left) and output characteristics (right) of a single device in the active-matrix array. **e** On-current mapping of a 30% biaxially stretched active-matrix array. **f** C-AFM image of a semiconducting film at 30% biaxial strain. **g** On-current mapping of a healed active-matrix array with cross-shaped damage. **h** C-AFM image of a healed semiconducting film with a cutting line width of 3 μm. Photographs and circuit diagrams of stretchable and self-healing logic devices: (**i**), inverter, (**j**), NAND, and (**k**), NOR. **l** Voltage–transfer characteristics of an inverter at 0% (top-left), 30% biaxial strain (top-right), after cutting (bottom-left), and after healing (bottom-right). The inset illustrations show the drive and load transistors consisting of inverter devices at pristine (top-left) and under 30% biaxial strain (top-right). The drive transistor is cut using surgical blade (bottom-left) and it is autonomously healed (bottom-right). Output curves at 0% (top-left), 30% biaxial strain (top-right), after cutting (bottom-left), and after healing (bottom-right) from (**m**), NAND and (**n**), NOR logic devices. The width and depth of cuts on all of the drive transistors were ~ 4 μm and 2 μm, respectively.

(Supplementary Fig. 57)[39]. Despite these morphological and electrical scars, the transistor maintained stable on/off switching behavior for up to 10,000 cycles even after healing (Supplementary Fig. 58). Additionally, the device reoperated even after electrical breakdown, although the electrically burned morphology of the device was not restored (Supplementary Fig. 59). Our stretchable self-healing transistors in the array also exhibited excellent ambient stability without encapsulation over 1 year (Supplementary Fig. 60a, b). We attributed the stability of the electrodes in air to self-encapsulation by supramolecular polymer in the semiconducting blend film (Fig. 1h and Supplementary Fig. 60c, d).

## Self-healable skin electronics

Last, we demonstrate the self-healing active-matrix arrays (Fig. 5a, b and Supplementary Fig. 61). The pristine array exhibited a uniform on-current in 25 devices, with a narrow deviation (Fig. 5c); different batches ($n = 3$) showed similar uniformity (Supplementary Fig. 62). The typical transfer and output characteristics of the representative unit device (Fig. 5d) showed low gate currents and operation at low voltage ($-1$ $V_D$). They maintained the initial on-currents under biaxial strain as high as 30% (Fig. 5e and Supplementary Fig. 63). Conductive-AFM images showed the current mapping of the active region of the semiconducting film at 30% biaxial and uniaxial strains (Fig. 5f and

Supplementary Fig. 64). The nanoweb-structured self-healing semiconductor was evenly stretched uniaxially and biaxially without electrical disconnections. This observation is direct evidence that the semiconducting film maintained the electrical percolation path under a biaxial strain (30%) and is suitable for applications on dynamic soft surfaces, such as human skin or organs. In addition, the array was tested for cross-shaped damage by deep cutting with a surgical blade to simulate harsh damage conditions. After 48 h, the typical electrical properties of all the damaged locations were restored, as demonstrated by the recovered on-current shown in Fig. 5g and Supplementary Fig. 65. The C-AFM image in Fig. 5h shows obvious reconnections of the conduction pathways after healing.

To demonstrate practical applications for our self-healing supramolecular polymer transistors, we prepared autonomously self-healable and stretchable inverters, NAND gates, and NOR gates, which are basic building blocks of digital integrated circuits (Fig. 5i–n and Supplementary Figs. 66, 67). The pristine inverter exhibited a typical voltage transfer curve (VTC) with a gain of 3, depending on the $V_{IN}$ and $V_{OUT}$ values, enabling 2-bit calculation. The VTC performance was similar to that of the pristine transistors under biaxial strain as high as 30% (Fig. 5l, top-right; Supplementary Figs. 68, 69). For self-healing experiments, the inverter exhibited similar performance as the pristine device after cut and heal at room temperature for 40 h, displaying negligible variation (Fig. 5l, bottom; Supplementary Fig. 70). In addition, the self-healed inverter was capable of stretching under biaxial strain as high as 30% without electrical disconnection (Supplementary Fig. 71a, b).

NAND and NOR gates were also fabricated with the self-healing transistors (Fig. 5j, k and Supplementary Figs. 72, 73). For output characteristics of the NAND and NOR devices (Fig. 5m, n, Supplementary Figs. 74–77) in their pristine state, $V_{OUT}$ exhibits a certain value according to the logic table and a similar trend under biaxial strain (Fig. 5m, n, top-right). When the two-drive transistor suffered damage (Supplementary Figs. 78, 79), the $V_{OUT}$ value became zero (Fig. 5m, n, bottom-left). After autonomous self-healing, $V_{OUT}$ returned to its original value (Fig. 5m, n, bottom-right), and the healed logic gates could stretch up to 30% biaxial strain again (Supplementary Fig. 71c, d).

We demonstrated autonomous self-healing transistors and circuit elements consisting of identical supramolecular polymer-based electronic components for skin electronics. Nanoweb-structured semiconductor network and stretchable metallization were developed for autonomous self-healing stretchable semiconductors and electrodes (source/drain and gate), respectively. In addition, the insulating high-k supramolecular elastomer was used for the low operating voltage of the transistor. The transistors showed strain-insensitive electrical properties even after autonomous self-healing. With the autonomous self-healing transistors, we successfully fabricated active-matrix array and logic gates that are fundamental building blocks of digital systems. These results would pave the way for the skin electronics based on integrated circuits with self-healing functional systems.

## Methods

### Materials
Bis(3-aminopropyl) terminated poly(dimethylsiloxane) ($H_2N$-PDMS-$NH_2$, $M_n = 5000$) was purchased from Gelest Inc.. Poly(2,5-bis(2-octyl-dodecyl)−3,6-di(thiophen-2-yl)diketopyrrolo[3,4-c]pyrrole-1,4-dione-alt-thieno[3,2-b]thiophen) (DPPT-TT) ($M_w/M_n = 100k/42k$, PDI < 3.0) semiconducting polymer was purchased from Derthon. Poly(dimethylsiloxane) (PDMS, Sylgard 184) and its cross-linker were purchased by Dow Corning. The PDMS was cured with a ratio of 8:1 (base/cross-linker, w/w) at 65 °C overnight for the transfer printing stamp. SEBS H1062 (S/EB weight ratio of 18/82) was supported by Asahi Kasei company. Ag (silver, 99.99%, 3–5 mm granule) was purchased from SY

SCIENCE. The Trichloro(octadecyl)silane (OTS) solution and diiodomethane were purchased by Sigma Aldrich. The Ethyl-amine ($Et_3N$), anhydrous chloroform, and toluene were purchased by Sigma Aldrich. Methanol (MeOH) solution was purchased by SAMCHUN. The all chemicals and materials were used without any purification.

### Synthesis of self-healing elastomer (SHE) PDMS-MPU$_{0.6}$-IU$_{0.4}$
PDMS-MPU$_{0.6}$-IU$_{0.4}$ polymer was synthesized according to previously reported methods[23]. $H_2NPDMS$-$NH_2$ (30 g, $M_n = 5000$, 1 eq) was dissolved in 120 mL Chloroform at 0 °C under nitrogen atmosphere. $Et_3N$ (3 mL) was added to the solution of $H_2NPDMS$-$NH_2$ and stirred for 1 h. After that, a mixture solution of 4,4′– Methylenebis(phenyl isocyanate) (MPU) (0.9 g, 0.6 eq) and Isophorone diisocyanate (IU) (0.54 g, 0.4 eq) in $CHCl_3$ was added dropwise. The solution was then allowed to warm to room temperature and stirred for 4 days.

After reaction, MeOH (5 mL) was added and stirred for 30 min to remove the remained isocyanate. Then, the solution was concentrated to ½ of its volume. 18 mL MeOH was poured into the mixture solution to give a viscous liquid. After settling for 30 min, the upper clear solution was then decanted. 30 mL $CHCl_3$ was added to dissolve the product. The dissolution-precipitation-decantation process was repeated three times for purifying and the final product was subjected in ambient condition to remove the solvent and trace of $Et_3N$.

### Device fabrication
**Rigid substrate-based thin film transistors.** The semiconductor solution was prepared by dissolving both DPPT-TT (0.21 wt%) and SHE (0.49 wt%) with a total of 0.7 wt%, in anhydrous chloroform at 50 °C for 4 h. The solution was spun on an OTS-treated $SiO_2$/Si wafer at 1000 rpm for 1 min after filtration with a PTFE-D (0.2 μm) filter (film thickness: 100 nm). The semiconducting film was then annealed at 80 °C for 30 min. All of above processes were carried out under an $N_2$ atmosphere in a glove box with extremely low levels of moisture ($H_2O < 0.01$ parts per million (ppm)) and oxygen ($O_2 < 0.01$ ppm).

The SHE dielectric solution (60 mg/mL in chloroform) was spun at 2000 rpm for 1 min (film thickness: 1.5 μm), without any heat treatment, onto an ITO glass substrate with a sheet resistance of 20 ohm/square. The semiconducting film was transferred onto the SHE dielectric using a PDMS stamp. The silver source/drain electrodes were then evaporated at a rate of 0.2 nm/s using a thermal evaporator. The channel length and width were set at 1000 and 150 μm, respectively. For calculation of effective Schottky barrier height, the transistor device was measured with structure (Ag/semiconductor/$SiO_2$/Si) in vacuum state.

**Fully stretchable and self-healing transistors.** The SHE substrate solution was spin-coated onto an OTS-treated $SiO_2$/Si wafer using the SHE solution (50 mg/mL in chloroform) at 2000 rpm for 1 min without any heat treatment. The resulting SHE substrate was transferred onto the SEBS substrate to be stretchability and elasticity. An 80 nm thick Ag gate electrode was evaporated onto the SHE substrate at a speed of 0.2 nm/s under high vacuum conditions (below $5.0 \times 10^{-6}$ torr). Next, the SHE dielectric (60 mg/ml, 2000 rpm for 1 min on OTS-treated $SiO_2$) and semiconducting film (in a 3:7 ratio of DPPT-TT and SHE) were sequentially transferred onto the gate electrode. Finally, 80 nm thick Ag source/drain electrodes were evaporated onto the semiconducting films.

**Active-matrix transistor array.** All the procedures were identical to the fabrication process of fully stretchable and autonomous self-healing transistors, except for the dielectric film. For the dielectric film, a different concentration (80 mg/ml in chloroform, 2.1 μm thickness) was used to reduce leakage current, and it was spin-coated at 1000 rpm for 1 min to achieve a thicker dielectric film.

**Logic gate devices including Inverter, NAND, NOR.** The self-healing substrate, semiconducting thin-film, and dielectric layer were prepared using the same process as the fully stretchable and self-healing active-matrix transistor array. Bottom Ag electrodes (80 nm) were evaporated onto the self-healing substrates, each designed for specific logic gates. The semiconducting films and dielectric layer were then transferred onto the Ag electrode in sequence. Finally, the top Ag electrodes (80 nm) were thermally evaporated following the designed pattern.

## Characterization

The electrical characteristics of the devices were measured using a four probe (MCP-T610) and probe station connected with KEITHLEY 4200 under ambient conditions. The capacitances of the dielectric were measured using a probe station connected with an LCR meter (Keysight 4274A). The strain-stress curve was obtained by a force tester (AND, MCT-2150; strain rate: 100 mm/min). For cutting process, a surgical blade (Feather, No.25) was used. UV-Vis-Nir spectra were obtained with a spectrophotometer (JASCO, V-770). Surface structures, and current images were obtained with atomic force microscopy (AFM; Bruker MultiMode 8-HR) under ambient conditions. DMT modulus mappings were measured using PeakForce quantitative nanoscale mechanical (QNM) AFM. Optical images were obtained with an optical microscope (OM; Leica DM4 M). The thicknesses of the film were obtained with an ellipsometer (WONWOO STRC-2000). Grazing incidence X-ray diffraction (GIXD) patterns of self-healing semiconducting thin films were performed on a laboratory beamline system (Xenocs Inc. Xeuss 2.0) with an X-ray wavelength of 1.54 Å and a sample-to-detector distance of 15 cm and the incidence angle of 0.2˚. Samples were kept under vacuum to minimize air scattering. Diffraction images were recorded on a Pilatus 1 M detector (Dectris Inc.) with an exposure time of 1.5 h and processed using the Nika software package, in combination with WAXSTolls in Igor Pro. X-ray photoelectron spectra depth profiling was obtained with XPS equipment (Thermo Electron, K-Alpha). Surface energy and contact angle were obtained by PHOENIX-MT(T).

## Data availability

Data are available on request. Correspondence and requests for materials should be addressed to T.I.L., Z.B., and J.Y.O.

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

## Acknowledgements

This work was supported by the National Research Foundation of Korea (NRF) (Grant No. 2021R1C1C1009925, Grant No. 2020R1A6A1A03048004, and Grant No. 2019R1A6C101052), Samsung Electronics Global Research Program, the GRRC program of Gyeonggi province (GRRCKYUN-GHEE2023-B03), and R&D program of the Ministry of Trade, Industry & Energy (No. 20015898, and No. 20012710) funded by the Korea Evaluation Institute of Industrial Technology (KEIT). This work was also supported by the Korea Institute for Advanced of Technology (KIAT) and the Ministry of Trade, Industry & Energy (MOTIE) of the Republic of Korea (No. P0017363 and No. 20019105). Part of this work was performed at the Stanford Nano Shared Facilities (SNSF), supported by the National Science Foundation under award ECCS-2026822. X.G. acknowledges National Science Foundation with grant number of DMR-2047689.

## Author contributions

Z.B. and J.Y.O. conceived the study and N.T.P.V., T.U.N., T.I.L., Z.B., and J.Y.O. designed the experiments. N.T.P.V. and T.U.N. conducted all experiments. N.T.P.V., T.U.N., M.W.J., J.S.K., K.H.J., Y.L., G.M., X.G., J.B.-H.T., T.I.L., Z.B., and J.Y.O. analyzed and discussed the data. N.T.P.V., T.U.N., T.I.L., J.B.-H.T., Z.B., and J.Y.O. wrote the manuscript.

## Competing interests

The authors declare no competing interests.
