## [Peer Review File · Nature Communications]

REVIEWER COMMENTS

Reviewer #1 (Remarks to the Author):

This well-detailed paper discusses a fully self-healing transistor - i.e., the semiconductor, metal, and insulator have all been given the attributes of healing and recovering from a mechanical fracture (or cut). While past research has studied the self-healing of one of the above constituents, this work has addressed all three. Therein lies its strength. In addition, the work is very detailed from the materials and methods aspect. Yet, what I felt was difficult to glean and truly comprehend from these details was a feel for the physics of healing and the reason for various aspects of device and circuit performance. Hence, all my queries are centered around this.

1. Is the blending of DPPT-TT in the PDMS (the SHE) followed by a curing process? It appears not.

(a) Yet, If yes: does the DPPT-TT phase separate into the nanofibers before curing or after?

(b) If not: what are the mechanical properties of the blend - in particular, the viscoelastic parameters? While the authors have shown the Young's Modulus, it would be more useful to look at the viscosity and the ability of the blend to flow, as this directly affects the time constant for healing. Further, is there a lifetime for the entire transistor? The flow would continue and result in a final deformation of the metal-insulator-semiconductor stack.

(The Methods section on the development of the SHE states that it remains a viscous liquid till the decantation process.)

I am trying to get a sense of what the device looks like. Is it a proper solid that never flows? Or is it a "gummy, gluey" gel that is reasonably stable - and flows when cut to cause healing? I get the perception that it is the latter.

2. Line 126: the physics of healing appears to be the flow of the material to fill in the cut. This once again returns to 1(b), the viscosity. Further, even after the material flows, the web-like fibers due to pi-pi interactions of DPPT-TT will have to be restructured to restore the semiconducting properties - what is the order of this time constant like? The total time for healing is of the order of a couple of days.

(Finally, if the answer to 1(a) is yes - i.e. if there is a curing process after blending, it should take a lot longer to reform this web?)

3. The optimized ratio of 3:7 offers a "balanced" Young's Modulus and a strain to generate a crack, i.e. the material is "flowy" enough and yet "solid" enough. It is also the point beyond which mobility degrades. This appears to be more than just a coincidence. Why does this happen? The mobility measurement here appears to be from the drain current due to field effect, and therefore the percolation pathway

matters. The 3:7 ratio is therefore defining a threshold beyond which the path from drain to source degrades. Why? Is there an obvious reason?

4. For field effect transistor operation, the region that would carry most current due to gating would be the semiconductor "webbing" plane closest to the dielectric layer. The rest of the network probably adds to leakage and not transistor action. Therefore, when the semiconductor is cut, the transistor action should be restored if the network plane closest to the gate is re-established. Once the blend flows in to fill the cut, the 36-hour point of Fig. 2g and 2i shows the leakage current drop before recovering at the 72-hour mark - seemingly corroborating the above argument i.e., the lower layer heals before the upper. Yet, the 12-hour point does not - it shows increased leakage and weaker transistor action - i.e. indicative of the fact that upper layers heal before the lower. Comments?

5. For self-healing electrodes - Ag complexes with the SHE have been formed using AgO to enable low contact resistance. Further, the barrier height for this interface with the semiconductor is shown in Fig. 3 to be about 0.1 eV. For self-healing transistors, the output characteristics have not been shown. (a) Was there any impact of the healing mechanism on this contact resistance? (b) Any impact of the self-healing mechanism on the channel length modulation resistance?

6 A broad question: The healing mechanism (for semiconductor, dielectric, or metal) is based on material flow from one region to another.

(a) Does this flow affect the electrical properties of the material in the regions that have lost mass i.e. from the regions where the materials flowed? Here are several thoughts: (i) repeated flow can cause undulations in the material (as seen by all profile scans). If this occurs at the semiconductor-insulator interface, it can cause poor transistor action (ii) the flow in the dielectric could reduce thickness and increase leakage (iii) the semiconductor flow implies that the DPPT-TT has to reorganize its filament network throughout (or live with strain in this network). This should affect performance in some manner. So, is there a limit on the number of cuts before such minor effects build up to a point of no return?

(b) temperature would be so critical for this. If the temperature is large, we could expect quicker healing. Yet, is there a critical temperature beyond which something would fail? Again, some questions: (i) would the whole transistor with the 3:7 ratio flow out and disintegrate at high temperatures? (ii) The mechanical and electrical properties of the semiconductor network would be strongly temperature dependent. Does increasing temperature disrupt the blend chemistry? Again, what is the point of no return regarding temperature?

(c) Further, if the 3:7 semiconductor can flow - it really wouldn't experience a mechanical "fracture" in a realistic application. Therefore, while the flow does offer healing, and demonstrations have been made with induced cuts, such a fault is not expected to happen due to natural mechanical causes. This leads to looking at other means of failure - Would the mechanism work if the semiconductor experiences mechanical degradation due to a forced high current for example?

Overall, I enjoyed reading this paper. The circuit demonstrations are impressive, and I would like to see this eventually published. However, I would like to get a better sense of the physics - mechanics and device physics, and the answers to these questions would help out.

Reviewer #2 (Remarks to the Author):

In this paper, the authors report on stretchable polymer transistors capable of autonomous self-healing. The authors have realized blends of electrically insulating supramolecular polymers with semiconducting polymers or deposited metal nanoclusters as active materials for the transistors. An important feature proposed by the authors is that by employing the same supramolecular self-healing polymer matrix for all active layers (conductor/semiconductor/dielectric layers) of the skinned transistor, adhesion and intimate contact between the layers is achieved, facilitating effective charge injection and transport under strain after self-healing. The authors have successfully fabricated skin-like self-healing circuits containing NAND and NOR gates and inverters, which are key components of arithmetic logic units, with the aim of demonstrating the utility of these transistors. The paper is well organized. I would like to recommend that the content of this paper be published because of its importance for future robust flexible electronics, but before doing so, let me review a few points.

1. Although the authors present excellent transistor characteristics, it is clear that the drive voltage is high considering a more practical point of view. I would like to request a more detailed introduction as to how low voltage drive is possible. In particular, I would like to see more detailed explanations of transistor characteristics, self-healing capability, and characteristic variation when the insulating film is made thinner. I would also like to see a strategy for low voltage drive.

In addition, I would like to see output characteristics displayed from the perspective of evaluating the magnitude of the injection barrier from the electrode to the channel when driving at low voltages.

2. Please explain hysteresis in transistor characteristics. I believe hysteresis can be seen in the transfer curve. Please show the relationship between the sweep speed of the voltage during measurement and hysteresis. Also, is the frequency response the same before and after self-healing? I can imagine damaged areas of the polymer becoming carrier trap sites, etc. Can the authors tell us if the effect of such damage shows up in the frequency response performance?

Overall, the paper is very well written and my questions are minor details to improve the paper. I would like to see it published with the above minor considerations added.

Reviewer 1

Overall evaluation: This well-detailed paper discusses a fully self-healing transistor - i.e., the semiconductor, metal, and insulator have all been given the attributes of healing and recovering from a mechanical fracture (or cut). While past research has studied the self-healing of one of the above constituents, this work has addressed all three. Therein lies its strength. In addition, the work is very detailed from the materials and methods aspect. Yet, what I felt was difficult to glean and truly comprehend from these details was a feel for the physics of healing and the reason for various aspects of device and circuit performance. Hence, all my queries are centered around this.

Our response: Thanks for your valuable comments on our work. All authors tried to fully address all of your comments with additional experiments as follows.

Comment #1. Is the blending of DPPT-TT in the PDMS (the SHE) followed by a curing process? It appears not. (a) Yet, If yes: does the DPPT-TT phase separate into the nanofibers before curing or after? (b) If not: what are the mechanical properties of the blend - in particular, the viscoelastic parameters? While the authors have shown the Young's Modulus, it would be more useful to look at the viscosity and the ability of the blend to flow, as this directly affects the time constant for healing. Further, is there a lifetime for the entire transistor? The flow would continue and result in a final deformation of the metal-insulator-semiconductor stack. (The Methods section on the development of the SHE states that it remains a viscous liquid till the decantation process.) I am trying to get a sense of what the device looks like. Is it a proper solid that never flows? Or is it a "gummy, gluey" gel that is reasonably stable - and flows when cut to cause healing? I get the perception that it is the latter.

Our response: We appreciate your constructive comment. The blend film (DPPT-TT:SHE) was thermally annealed at 80 °C for 30 min after being spin-coated onto the substrate as a post treatment to evaporate residual solvent and optimize molecular structure, not for chemical crosslinking. The AFM images of pristine and thermally annealed films are presented in **Figure R1**, and we found the nanoweb structure in the pristine film and the morphologies were identical after the thermal annealing process.

To gain insight into the mechanical properties of the blend film, we conducted dynamic mechanical analysis (DMA) on the blend film (DPPT-TT:SHE, 3:7). **Figure R2a** shows the storage and loss moduli of the blend film as a function of frequency at 35 °C (near room temperature). Across all frequency ranges from 7×10^{-1} Hz to 1.2×10^1 Hz, the storage modulus consistently exceeded the loss modulus, indicating the solid-like behavior of the blend film ($\tan \delta < 1$). Additionally, as the frequency increased, the storage modulus showed a gradual rise, while the loss modulus decreased, suggesting viscoelastic property in the blend film.

Furthermore, we investigated the storage and loss moduli of the blend film as a function of temperature (ranging from -20 °C to 60 °C at 1 Hz). Remarkably, the storage modulus maintained a higher value than the loss modulus across the entire temperature range (**Figure R2b**). To assess macroscopic flow behavior, we measured the change in length of the bulk blend film suspended under gravity for 7 days (inset in **Figure R3**), and no change in the length of the bulk blend film was observed.

Regarding the lifetime of our self-healing transistor, we initially measured the conductivity of the Ag electrode and the field-effect mobility of the transistor and on October 17, 2022 and November 22, 2022, as shown in **Figure R4, R5**, respectively. Interestingly, when we re-measured the same transistor on January 22, 2024, after over 1 year, it almost maintained its initial mobility values, along with the conductivity of the electrodes, despite the absence of a passivation layer. These devices were stored in a desiccator, maintaining low relative humidity (< 20%) and room temperature in ambient air. This result is of significant interest to all the authors during the revision process.

[Added text in the revised manuscript]

1. On page 5 for Figure R1 (Supplementary Fig. 2): The nanoweb morphology was observed in as-spun blend film, which was a result of phase separation due to the difference in surface energy between the materials, and the nanoweb morphology in the SHE matrix maintained after thermal annealing at 80 °C for 30 min (Supplementary Figs. 1 and 2).

2. On page 5 for Figure R2, R3 (Supplementary Figs. 6 and 7): To gain insight into the mechanical properties of the 3:7 blend film, dynamic mechanical analysis (DMA) was conducted on the blend film (DPPT-TT:SHE, 3:7). The storage modulus (G') consistently exceeded the loss modulus (G'') across the 10^{-1} to 10^1 Hz frequency range at 35°C, and the difference between the storage and loss moduli became larger as the frequency increased. This indicates that the blend film exhibits solid-like viscoelastic properties ($\tan \delta = G'/G'' < 1$) in room temperature (Supplementary Fig. 6a). In addition, the temperature dependency of the $\tan \delta$ was investigated from -20 °C to 60 °C at 1 Hz and the $\tan \delta$ remained below 1 in the temperature range, which means that the blend film behaves like a solid-like material (Supplementary Fig. 6b). The unchanged length of the blend film suspended under gravity for 7 days at room temperature supports its solid-like behavior (Supplementary Fig. 7).

3. On page 14 for Figure R4, R5: Our stretchable self-healing transistors in the array also exhibited excellent ambient stability without encapsulation over 1 year (Extended Data Fig. 9).

Figure R1 (Supplementary Fig. 2). AFM images (left: height, right: phase) of the blend film (3:7 weight ratio, DPPT-TT:SHE) **a**, before and **b**, before (top) and after (bottom) thermal annealing at 80 °C for 30 min.

Figure R2 (Supplementary Fig. 6). Dynamic mechanical properties of the blend film (3:7, DPPT-TT:SHE). Storage modulus (G'), loss modulus (G'') and $\tan \delta$ (G''/G') according to **a**, frequency and **b**, temperature, respectively.

Figure R3 (Supplementary Fig. 7). Photographs and length change of DPPT-TT:SHE film suspended under gravity for 7 days at room temperature.

Figure R4 (Extended data Fig. 9c,d). Resistance changes of **a**, S/D electrode (silver on semiconductor) and **b**, gate electrode (silver on substrate) in ambient air condition.

Figure R5 (Extended data Fig. 9a,b). Normalized field-effect mobility changes in ambient air condition: **a**, without encapsulation and **b**, with SHE encapsulation.

Comment #2. Line 126: the physics of healing appears to be the flow of the material to fill in the cut. This once again returns to 1(b), the viscosity. Further, even after the material flows, the web-like fibers due to pi-pi interactions of DPPT-TT will have to be restructured to restore the semiconducting properties - what is the order of this time constant like? The total time for healing is of the order of a couple of days. (Finally, if the answer to 1(a) is yes - i.e. if there is a curing process after blending, it should take a lot longer to reform this web?)

Our response: Thank you for your constructive comments. We observed the self-healing process of the damaged film (3.5 μm cutting) in micro- and nanoscales using OM and AFM equipment, respectively. Dark-field OM can track the damaged region through transmitted light from bottom to top microscope lens at the cutting line (**Figure R6**). Consequently, we can confirm the time when the cutting line completely disappeared in microscale. **Figure R6b** shows the cutting line completely disappeared after 12 hours. To gain insight into the healing process of the blend film at the nanoscale level, we conducted AFM height measurements during cutting and subsequent healing stages (pristine, cutting, 20 h healing, and 40 h healing), as depicted in **Figure R7a-d**. We observed that the nanoscale valley formed around the cutting line became filled with the blend materials and flattened over 40 hours (healing time). Line profiling of the AFM height images clearly illustrates the healing progress of the damaged region of the blend film. Moreover, the damaged network of the semiconducting nanoweb was nearly reconstructed after 40 hours of healing. This reconstruction can be attributed primarily to its viscoelasticity, which possesses partially viscous properties. This allows the material to fill the damaged region (valley) while relaxing the applied strain caused by cutting (see **Figure R2**).

Figure R6 (Supplementary Fig. 18). OM images of the damaged blend film (damage width: 3.5 μm) as a function of healing time at room temperature; **a**, bright field images and **b**, dark field images. Scale bar: 50 μm .

Figure R7 (Extended Data Fig. 3). AFM height images of the DPPT-TT:SHE films for self-healing process: **a**, pristine **b**, after cut, **c**, after 20 hours and **d**, after 40 hours states. Line profiling of the AFM height image **e**, pristine, **f**, after cut, **g**, after 20 h and **h**, after 40 h states. As time flows, the cutting line was healed without any treatment.

Comment #3. The optimized ratio of 3:7 offers a "balanced" Young's Modulus and a strain to generate a crack, i.e. the material is "flowy" enough and yet "solid" enough. It is also the point beyond which mobility degrades. This appears to be more than just a coincidence. Why does this happen? The mobility measurement here appears to be from the drain current due to field effect, and therefore the percolation pathway matters. The 3:7 ratio is therefore defining a threshold beyond which the path from drain to source degrades. Why? Is there an obvious reason?

Our response: Thank you for your constructive comments. First of all, we are sorry for confusing the precondition of the optimal blend ratio for the self-healing semiconductor film. We considered four factors (1. Morphology, 2. Field-effect mobility, 3 Crack-on-set strain and 4. Young's modulus) for the optimal blend ratio and would like to explain the reason why we chose the 3:7 (DPPT-TT:SHE) blend ratio. **Figure R8** shows the AFM images of the blend film as a function of blend film (from 1:9 to 9:1, DPPT-TT:SHE). The nanofibers of DPPT-TT formed the nanoweb structure in the SHE matrix. In case of higher DPPT-TT content than 3:7 film (9:1, 8:2, 7:3, 5:5 > 3:7, DPPT-TT:SHE), there were DPPT-TT nanofibers in the blend film. However, the lower SHE contents led to lower crack-on-set strain and higher Young's modulus, which causes strain-sensitive electrical property, although they had slightly higher field-effect mobility compared to the 3:7 as shown in **Figure R9**. In contrast, in case of lower DPPT-TT content than 3:7 blend film (1:9, 2:8 < 3:7, DPPT-TT:SHE), the nanoweb structures of DPPT-TT were disconnected as shown in **Figure R8** although these had lower Young's modulus and higher crack-on-set strain than 3:7 blend ratio, which effected the electrical properties of the blend film on strain (**Figure R9**, in case of a transistor with 1:9 blend ratio, the device didn't work).

[Added text in the revised manuscript]

1. On page 6 for response to comment #3: Consequently, the four kinds of material factors (nanoweb morphology, strain-insensitive electrical property, crack-on-set strain, and Young's modulus) were considered for the optimal blend ratio of the semiconducting film.

Figure R8 (Extended Data Fig. 1). AFM height images of the blend films with various weight ratios (DPPT-TT:SHE, 9:1 to 1:9).

Figure R9 (Extended Data Fig. 2). Transfer characteristics of films with various blend ratios (DPPT-TT:SHE) under 0% to 30% uniaxial strain.

Comment #4. For field effect transistor operation, the region that would carry most current due to gating would be the semiconductor "webbing" plane closest to the dielectric layer. The rest of the network probably adds to leakage and not transistor action. Therefore, when the semiconductor is cut, the transistor action should be restored if the network plane closest to the gate is re-established. Once the blend flows in to fill the cut, the 36-hour point of Fig. 2g and 2i shows the leakage current drop before recovering at the 72-hour mark - seemingly corroborating the above argument i.e., the lower layer heals before the upper. Yet, the 12-hour point does not - it shows increased leakage and weaker transistor action - i.e. indicative of the fact that upper layers heal before the lower. Comments?

Our response: Thank you for your insightful comment. The explanation for the changes in the I-V characteristics during the healing process is as follows. Initially, depicted by the red dashed line in **Figure R10**, the device loses transistor behavior due to the cut of semiconductor web and a significant leakage current occurs through the created free surface in the dielectric layer.

Subsequently, the device undergoes a gradual self-healing process over time, as noted by the reviewer, with recovery occurring in the sequence of the dielectric, the lower semiconductor web, and the upper semiconductor web. In **Figure R10** at 12 hours, there is a recovery in the lower semiconductor web, resulting in weak transistor behavior operating in enhancement mode with a low carrier concentration. However, since the dielectric recovery is insufficient, it still exhibits a high off-current. After 36 hours, the dielectric recovery is complete, leading to a very low off current. Over time, the upper semiconductor web's recovery takes place, causing an increase in carrier concentration, a positive shift of V_{th} , and a rise in both off- and on-currents.

Figure R10 (Figure 2g-j). **g**, Transfer characteristics ($V_D = -60$ V) with a cutting line ($3 \mu\text{m}$ width) perpendicular to the channel as a function of the healing time. **h**, Normalized field-effect mobilities (μ_0 : pristine mobility) with different cutting widths for transistors subjected to cutting perpendicular to the channel direction as a function of healing time ($n = 3$). **i**, Transfer characteristics ($V_D = -60$ V) with a cutting line ($3 \mu\text{m}$) parallel to the channel as a function of healing time. **j**, Normalized field-effect mobilities (μ_0 : pristine mobility) with different cutting widths parallel to the channel direction as a function of the healing time ($n = 3$).

Comment #5. For self-healing electrodes - Ag complexes with the SHE have been formed using AgO to enable low contact resistance. Further, the barrier height for this interface with the semiconductor is shown in Fig. 3 to be about 0.1 eV. For self-healing transistors, the output characteristics have not been shown. (a) Was there any impact of the healing mechanism on this contact resistance? (b) Any impact of the self-healing mechanism on the channel length modulation resistance?

Our response: Thank you for your valuable comments. We added the output curves of the self-healing transistor before and after autonomous healing in **Figure R11**. The healed device exhibited a slight degradation in drain currents, however, the device showed distinct healing of current compared cut device.

To investigate the healing mechanism of channel length modulation, we measured the resistance changes in various channel lengths for pristine and healed devices (**Figure R12a**) by equation: $R_{On}W = R_{channel}W + R_{contact}W = \frac{L}{\mu_c C_i (V_G - V_T)} + R_c W$, where L is channel length, W is channel width using the transfer-length method (TLM). All devices were cut in the middle of the channel, and interestingly, all devices exhibited successful healing regardless of the channel length. Furthermore, the resistance values for each channel length were similar. Regarding the contact resistance, the healed transistor showed a slightly higher contact resistance compared to the pristine devices, possibly due to the partial scar regions that were not completely healed of the Ag electrodes (**Figure R12b**). Thus, the cutting of the channel does not seem to impact the contact-related issues.

[Added text in the revised manuscript]

1. On page 13 for Figure R11, R12 (Supplementary Figs. 47 and 48): However, all the transistors were able to autonomously self-heal over time (duration dependent on cut severity),

and eventually restored most of their original electrical properties and morphology (**Fig. 4g,h, Extended Data Fig. 8, Supplementary Figs. 46-48**).

Figure R11 (Supplementary Fig. 47). Output characteristics of **a**, pristine, **b**, cut, and **c**, healed transistors.

Figure R12 (Supplementary Fig. 48). **a**, Resistance change of self-healing transistor as a function of channel length ($V_D = -5$ V). **b**, Comparison of extracted contact resistance (R_C) with pristine and healed devices.

Comment #6. A broad question: The healing mechanism (for semiconductor, dielectric, or metal) is based on material flow from one region to another. **Comment #6(a).** (a) Does this flow affect the electrical properties of the material in the regions that have lost mass i.e. from the regions where the materials flowed? Here are several thoughts: (i) repeated flow can cause undulations in the material (as seen by all profile scans). If this occurs at the semiconductor-insulator interface, it can cause poor transistor action (ii) the flow in the dielectric could reduce thickness and increase leakage (iii) the semiconductor flow implies that the DPPT-TT has to reorganize its filament network throughout (or live with strain in this network). This should affect performance in some manner. So, is there a limit on the number of cuts before such minor effects build up to a point of no return?

Our response: Thanks for your constructive comment. We totally agree with the three scenarios that the reviewer suggested. The material loss and the electrical performance of the healed device are closely related with damage shape. In this work, the very sharp knife (surgery blade) was used for clear-cutting to minimize material loss and material mixing at layer interfaces. In case of the dielectric, the capacitance value was almost preserved after healing in MIM structure (Ag/SHE/Ag), which indicates that there is negligible material loss without electrical breakdown and short as shown in **Figure R13 (Figure 4)**.

To investigate the reorganization behavior of the DPPT-TT nanoweb structure of the blend film at multiple cutting, the blend film was repeatedly cut for two times, and we observed the nanomorphology of the cut and healed blend films as shown in **Figure R14**. The disconnected DPPT-TT nanoweb structure was reorganized with partially disordered network at the first cutting. We cut the healed blend film at the same region again (second cut) and found that further disordered DPPT-TT network after second healing process, which implies that the self-healing of the blend film is repeatable, but the reorganized DPPT-TT nanoweb of the healed blend film becomes disorder.

[Added text in the revised manuscript]

1. On page 8 for Figure R14 (Supplementary Fig. 21): The autonomous self-healing of the blend film was found to be repeatable, even if the healed region is damaged again (Supplementary Fig. 21).

Figure R13 (Figure 4). **a**, Current density versus electrical field plots of the self-healing dielectric in MIM structure. **b**, Capacitance and dielectric constant of a self-healing dielectric in its pristine and healed ($3 \mu m$ cutting width) states.

Figure R14 (Supplementary Fig. 21). AFM phase images of the blend film (3:7, DPPT-TT:SHE) during multiple cutting and healing at same region.

Comment #6(b). Temperature would be so critical for this. If the temperature is large, we could expect quicker healing. Yet, is there a critical temperature beyond which something would fail? Again, some questions: (i) would the whole transistor with the 3:7 ratio flow out and disintegrate at high temperatures? (ii) The mechanical and electrical properties of the semiconductor network would be strongly temperature dependent. Does increasing temperature disrupt the blend chemistry? Again, what is the point of no return regarding temperature?

Our response: Thank you for your constructive comment. We conducted *in-situ* I-V transfer curve measurements of the self-healing transistor while varying the temperature from -20 °C to 140 °C in the nitrogen filled chamber (**Figure R15a**). As the temperature increased up to 80 °C, the on-current of the transistor gradually increased due to its hopping transport mechanism. However, the transfer curve of the transistor abruptly degraded around 100 °C, and the device lost its switching behavior at 140 °C, which is attributed to the low melting point of the self-healing elastomer. Consequently, the self-healing transistor ceased to function as a solid-state semiconductor device above 140 °C. Interestingly, after cooling to room temperature, the transistor was re-operated, although its electrical properties were degraded.

[Added text in the revised manuscript]

1. On page 12 for Figure R15 (Supplementary Fig. 40): The autonomous self-healing supramolecular polymer transistor showed temperature dependent on-current based on hopping transport and operated up to 100 °C while maintaining its switching behavior (Supplementary Fig. 40).

Figure R15 (Supplementary Fig. 40). **a**, Photograph of temperature dependence measurement. **b**, Transfer characteristics and **c**, mobility under various temperature conditions ($V_D = -60$ V). RT means cooling to room temperature from 140 °C. The scatter in Fig. R15b is the transfer curve for colling to RT condition. Inset of Fig R15c is the schematic of device structure and photographs of devices before and after temperature measurement.

Comment #6(c). Further, if the 3:7 semiconductor can flow - it really wouldn't experience a mechanical "fracture" in a realistic application. Therefore, while the flow does offer healing, and demonstrations have been made with induced cuts, such a fault is not expected to happen due to natural mechanical causes. This leads to looking at other means of failure - Would the mechanism work if the semiconductor experiences mechanical degradation due to a forced high current for example?

Our response: Thank you for your comment. As the reviewer suggested, we tested the self-healing ability of the transistor in an extreme case of electrical breakdown. **Figure R16a** shows an electrical breakdown transistor subjected to an applied voltage of -150 V_G and V_D . Unlike physical cutting, the morphology burned (point a) due to the electrical short did not recover even after 60 hours (**Figure R16b**). However, interestingly, the device began to work again when we connected the edge point (point b) of the source and drain electrodes near the burned region (**Figure R16c**).

[Added text in the revised manuscript]

1. On page 13 for Figure R16 (Supplementary Fig. 51): Additionally, the device reoperated even after electrical breakdown, although the electrically burned morphology of the device was not restored (Supplementary Fig.51).

Figure R16 (Supplementary Fig. 51). Photographs of **a**, an electrical breakdown transistor subjected to an applied voltage of $-150 V_G$ and V_D , and **b**, the same transistor after 24 hours. **c**, I-V transfer curves of the self-healing transistor after experiencing a breakdown, 24 and 60 hours later. The transfer characteristic at point b after 24 and 60 hours.

Comment #7. Overall, I enjoyed reading this paper. The circuit demonstrations are impressive, and I would like to see this eventually published. However, I would like to get a better sense of the physics - mechanics and device physics, and the answers to these questions would help out.

Our response: We sincerely appreciate Reviewer #1 for the positive and encouraging comments. We are sure that these comments help improve the quality of the manuscript significantly.

Reviewer 2

Overall evaluation: In this paper, the authors report on stretchable polymer transistors capable of autonomous self-healing. The authors have realized blends of electrically insulating supramolecular polymers with semiconducting polymers or deposited metal nanoclusters as active materials for the transistors. An important feature proposed by the authors is that by employing the same supramolecular self-healing polymer matrix for all active layers (conductor/semiconductor/dielectric layers) of the skinned transistor, adhesion and intimate contact between the layers is achieved, facilitating effective charge injection and transport under strain after self-healing. The authors have successfully fabricated skin-like self-healing circuits containing NAND and NOR gates and inverters, which are key components of arithmetic logic units, with the aim of demonstrating the utility of these transistors. The paper is well organized. I would like to recommend that the content of this paper be published because of its importance for future robust flexible electronics, but before doing so, let me review a few points.

Our response: Thanks for your valuable comments on our work. All authors tried to fully address all of your comments with additional experiments as follows.

Comment #1. Although the authors present excellent transistor characteristics, it is clear that the drive voltage is high considering a more practical point of view. I would like to request a more detailed introduction as to how low voltage drive is possible. In particular, I would like to see more detailed explanations of transistor characteristics, self-healing capability, and characteristic variation when the insulating film is made thinner. I would also like to see a strategy for low voltage drive. In addition, I would like to see output characteristics displayed from the perspective of evaluating the magnitude of the injection barrier from the electrode to the channel when driving at low voltages.

Our response: Thanks for your constructive comments. The drain voltage (V_D) and gate voltage (V_G) of our self-healing transistors can be reduced to -1V from -60 V as shown in Figure 5d due to the relatively high-k ($k: 7.2$) self-healing dielectric compared to conventional polymer dielectric materials, as shown in Figure 5d. However, as the reviewer suggested lowering the operating voltage of the transistor compared to current devices, we conducted additional experiments to reduce the thickness of the self-healing dielectric from 1.5 μm to 600 nm, which is an efficient way to increase the capacitance of the dielectric layer, following the simple equation: $C = k \cdot \epsilon_0 \cdot A/t$ (C : dielectric capacitance, ϵ_0 : space permittivity, k : dielectric constant, A : area, and t : thickness). **Figure R17a-d** show the transfer and output curves of the self-healing transistors with various dielectric thicknesses (1500 nm, 1200 nm, 900 nm, and 600 nm). Contrary to our expectations, the on-currents (drain currents, I_D) of the devices at the same V_G gradually decreased as we reduced the thickness of the dielectric layer. The abnormal trend in the devices can be explained by the increased gate leakage current, which is related to the leakage current of the transistor between the source-drain electrode and gate electrode, weakening the field effect induced by the gate voltage, even though the dielectric thickness

was reduced to obtain higher capacitance. Notably, in ultrathin dielectric thickness (600 nm), the failed operation as field effect transistor devices in high gate voltage region was occurred.

In addition, the thickness of the dielectric significantly affects the self-healing performance of the devices. **Figure R17e-h** show the transfer curves of the self-healing transistor with various dielectric thicknesses during a 48-hour self-healing process. In case of ultrathin dielectric (600 nm), to precisely investigate self-healing ability, we applied the low gate voltage (+2 to -15 V) to prevent the broken-down device. As the dielectric thickness decreases, the self-healing performance of the transistor deteriorates significantly, even though all the devices were operational again. Thinner dielectric layers have difficulty realigning all the component layers of the damaged (cut) transistor. Consequently, finding the optimized thickness of the self-healing dielectric layer is necessary to balance the trade-off between low-power operation and self-healing performance while minimizing leakage current.

On the other hand, reducing the thickness of the dielectric layer has less effect on the contact resistance and current injection barrier of the self-healing transistor. **Figure R18** shows the contact resistance and current injection barrier. All devices exhibited similar contact resistance and current injection barrier values, regardless of the thickness of the dielectric layer because both factors are closely related with the property of source electrode-semiconductor interface.

[Added text in the revised manuscript]

1. On page 11 for Figure R17, R18 (Supplementary Figs. 35 and 36): Although a thinner dielectric layer can lead to a lower operating voltage in transistors, dielectric layers with a thickness of less than 1.5 μm resulted in a lower breakdown voltage and higher leakage current in the devices (Supplementary Fig. 35a-d). Additionally, thinner dielectric layer faces a challenge in aligning healing areas at the submicron scale, which leads to a reduced self-healing ability, even though the contact resistance and current injection did not affect the devices. (Supplementary Figs. 35e-h and 36).

Figure R17 (Supplementary Fig. 35). a-d, Transfer and output curves of self-healing transistors with various dielectric thicknesses and different drain voltage. e-h, Transfer curves

of the self-healing transistors operating at $-5 V_D$ in -60 to $20 V_G$ while self-healing process for 48 h.

Figure R18 (Supplementary Fig. 36). a, Contact resistances and b-e, current injection barriers of the self-healing transistor as a function of dielectric thickness ranging from 1,500 nm to 600 nm. Inset of b-e are the output characteristics in low voltage operation.

Comment #2. Please explain hysteresis in transistor characteristics. I believe hysteresis can be seen in the transfer curve. Please show the relationship between the sweep speed of the voltage during measurement and hysteresis. Also, is the frequency response the same before and after self-healing? I can imagine damaged areas of the polymer becoming carrier trap sites, etc. Can the authors tell us if the effect of such damage shows up in the frequency response performance?

Our response: Thank you for your comments. We analyzed the transfer characteristics of the transistors under various voltage sweep speeds, as depicted in **Figure R19a**. Across all sweep speeds, we observed an anticlockwise hysteresis in the transfer curve, characterized by lower reverse currents compared to forward sweep currents. This behavior is attributed to charge carrier trapping near the channel and semiconductor-dielectric interface, as referenced³⁹. The degree of electrical hysteresis was quantified by the shift in threshold voltage (ΔV_{th}) and it depends on the I-V sweep speed as shown in **Figure R19b**, which is a typical behavior of the charge trap based electrical hysteresis. This behavior is distinct from the effects induced by mobile ion impurities or polarization of dielectric, which typically result in clockwise hysteresis (higher reverse sweep current compared to forward sweep current)³⁹.

To assess the change in frequency response of the device before and after healing, we measured the capacitance versus voltage (C-V) curves of the self-healing metal-insulator-semiconductor (MIS) capacitor across a range of frequencies from 0.1 kHz to 100 kHz. **Figure R20** illustrates the C-V curves of the MIS capacitor both before and after healing. Prior to healing, the MIS capacitor exhibited typical C-V curves depending on the frequency. However,

post-healing, we observed a slight hump in all the C-V curves of the transistor, particularly between 0 V to 10 V. This phenomenon indicates an increase in the number of charge trap sites localized at the semiconductor/dielectric interface⁴⁰.

Further investigation into the switching performance of device involved conducting a continuous on/off switching test up to 10,000 cycles, both before and after healing, as depicted in **Figure R21**. Remarkably, the device demonstrated stable on/off switching performance even after healing. This suggests that the presence of electrical scars in the healed region had minimal effect on the device's on/off switching behavior.

[Added text in the revised manuscript]

1. On page 12 for Figure R19 (Supplementary Fig. 39): The devices exhibited an anticlockwise hysteresis that is attributed to charge carrier trapping near the channel and semiconductor-dielectric interface³⁹. The degree of electrical hysteresis was quantified by the shift in threshold voltage (ΔV_{th}), which depends on the I-V sweep speed. (Supplementary Fig. 39).

2. On page 13 for Figure R20 (Supplementary Fig. 49): To assess the frequency response of the healed device, capacitance versus voltage (C-V) curves of the self-healing metal-insulator-semiconductor (MIS) capacitor were measured across different frequencies. The pristine MIS capacitor exhibited typical C-V curves varying with frequency. After healing, a slight hump was observed in the C-V curves between 0 V and 10 V, indicative of increased charge traps at the semiconductor/dielectric interface (Supplementary Fig. 49)⁴⁰.

3. On page 13 for Figure R21 (Supplementary Fig. 50): Despite these morphological and electrical scars, the transistor maintained stable on/off switching behavior for up to 10,000 cycles even after healing (Supplementary Fig. 50).

Figure R19 (Supplementary Fig. 39). **a**, Transfer characteristics of the self-healing transistor and **b**, threshold voltage shift (ΔV_{th}) as a function of sweeping speed of gate voltage (V_G).

Figure R20 (Supplementary Fig. 49). Capacitance versus voltage (C-V) curves as a function of frequency of the MIS capacitor in a, pristine and b, healed states. Inset in a is the schematic for MIS structure and the method to measure.

Figure R21 (Supplementary Fig. 50). Continuous on/off switching test with 10,000 cycles of the autonomous self-healing supramolecular polymer transistor **a**, before and **b**, after healing ($V_D = -10$ V, and $V_{GS} = -60$ V_{on}, +2 V_{off}).

Comment #3. Overall, the paper is very well written and my questions are minor details to improve the paper. I would like to see it published with the above minor considerations added.

Our response: We sincerely appreciate Reviewer #2 for the positive and encouraging comments. We are sure that these comments help improve the quality of the manuscript significantly the revised manuscript has truly benefited from your comments and suggestions.

References

39. Egginger, M., Bauer, S., Schwödiauer, R., Neugebauer, H., & Sariciftci, N. S. Current versus gate voltage hysteresis in organic field effect transistors. *Monatsh. Chem.* **140**, 735-750 (2009).
40. Castagne, R., & Vapaille, A. Description of the SiO₂ – Si interface properties by means of very low frequency MOS capacitance measurements. *Surf. Sci.* **28**, 157-193 (1971).

REVIEWERS' COMMENTS

Reviewer #1 (Remarks to the Author):

The authors have answered my queries.

Best wishes.

Reviewer #2 (Remarks to the Author):

The authors have carefully responded to the points raised by the reviewers and the paper has been revised appropriately. This content is very well organized and I would recommend it for publication.

Reviewer 1

Comment #1. The authors have answered my queries.

Our response: Thank you for your comment.

Reviewer 2

Comment #1. The authors have carefully responded to the points raised by the reviewers and the paper has been revised appropriately. This content is very well organized and I would recommend it for publication.

Our response: Thank you for your comment.